# Air Pollution (PM_2.5_) Negatively Affects Urban Livability in South Korea and China

**DOI:** 10.3390/ijerph192013049

**Published:** 2022-10-11

**Authors:** Sunmin Jun, Mengying Li, Juchul Jung

**Affiliations:** 1BK21PLUS, Department of Urban Planning and Engineering, Pusan National University, Busan 46241, Korea; 2Department of Urban Planning and Engineering, Pusan National University, Busan 46241, Korea

**Keywords:** vulnerability to PM_2.5_, living conditions, livability, grey relational analysis, transboundary, Korea, China

## Abstract

This study investigated the effect of the concentration of ambient fine particulate matter $\left({\mathrm{PM}}_{2.5}\right)$, a transboundary air pollutant, on the livability of neighboring areas of China and South Korea with the aim of informing common policy development. Grey relational analysis (GRA) and panel regression analysis were performed to examine the effect of ${\mathrm{PM}}_{2.5}$ concentration on various livability indicators. The results revealed that urban living infrastructure was an indicator of effect in both South Korea and China. Based on the high correlation between urban living infrastructure and ${\mathrm{PM}}_{2.5}$ concentration, it can be seen that ${\mathrm{PM}}_{2.5}$ clearly affects livability, shown by panel regression analysis. Other key livability indicators were traffic safety, culture and leisure, and climate indicators. Spatial analysis of the livability index revealed that from 2015 to 2019, livability improved in both South Korea and China, but there was a clear difference in the spatial distribution in China. High-vulnerability areas showed potential risks that can reduce livability in the long run. In South Korea and China, areas surrounding large cities were found to be highly vulnerable. The findings of this research can guide the establishment of policies grading ${\mathrm{PM}}_{2.5}$ pollution at the regional or city macro-level.

## 1. Introduction

According to the World Health Organization’s announcement in 2022, 99% of the world’s population lives in areas that do not meet the recommended air quality standards, and 70% of deaths from ambient fine particulate matter (${\mathrm{PM}}_{2.5}$) exposure occurred in East Asia and the Pacific, and South Asia. China and India accounted for 52% of global deaths from ${\mathrm{PM}}_{2.5}$, with South Korea having the lowest rate among Organization for Economic Cooperation and Development (OECD) member countries [1]. Air pollution is one of the greatest threats to human health and can impede economic growth by increasing medical expenses and lowering labor productivity. Furthermore, the increase in particulate matter lowers an individual’s happiness index [2,3,4,5]. As ${\mathrm{PM}}_{2.5}$ is a transboundary pollutant that moves beyond national borders, it damages not only the country concerned but also neighboring countries. It has been widely reported as an aggravating factor for the environment [6,7]. In particular, Korea is very close to the coastal areas of China, so there is a high concern about the impact of ${\mathrm{PM}}_{2.5}$. In order to reduce ${\mathrm{PM}}_{2.5}$, not only the efforts of individual countries but also transboundary consultations and joint responses with neighboring countries are required.

${\mathrm{PM}}_{2.5}$ has attracted substantial attention as a primary pollutant that plays a pivotal role in the deterioration of the quality of urban living environments [8]. The urban movement began when environmental pollution brought on by industrialization and urbanization caused problems and gave rise to the term “livability,” which refers to the quality of the urban living environment. Livability is the right of a citizen to live in appropriate living conditions [9], defined as “livable living conditions.” A natural environment is a prerequisite for health and is the most important and fundamental factor in livability [10]. Empirical studies have revealed that the environment is a major determinant of livability, and in particular, air quality has been found to have the greatest effect [11,12]. More recent studies have measured livability based on indicators of natural disasters such as abnormal climates [13], heat waves [14], and floods due to climate change [15]. Kim and Jin [5,16] developed a method to estimate the value of environmental goods using the happiness index to study how particulate matter affects the quality of life. 

The concept of livability emerged in the Netherlands in 1950 against the background of poor rural living conditions and refers to the human right to healthy living conditions. Since then, it has been used to guide the creation of cities where citizens can live comfortably, as well as to reduce environmental pollution and reckless urban development, and warn of the wide range of environmental effects of the latter. Consequently, livability has been highlighted as a new urban policy ideology that emphasizes democracy and active citizenship [9]. In other words, livability is considered the basis for the protection of citizens from noise and environmental pollution and their right to a safe environment, as well as for their responsibility. Livability is a measure of the quality of a person–environment relationship based on location [17]. Veenhoven [18] defined livability as “livable living conditions” and stated that it indicates that institutional arrangements meet human needs and capacities. Livability is an umbrella term for various environmental qualities and is measured as the sum of objective indicators that improve the quality of life, such as the economy, society, safety, transportation, and culture [15]. From the perspective of urban sustainability, it is important to reduce the environmental impact of rapid urbanization to ensure livability [19]. Negative indicators, such as environmental pollution, negatively affect people’s living burden and livability, thereby lowering their quality of life [20].

An important factor to consider in the measurement of livability is ${\mathrm{PM}}_{2.5}$ vulnerability because it is closely related to the population and the socio-ecological system, which are factors affecting livability [21]. Klinenberg [22] found through empirical studies that the death rate due to heat waves has a strong correlation with social vulnerabilities, such as racial segregation and inequality. The reason for the demographically similar but regionally different results is that the social infrastructure, i.e., physical space and group differences that determine how people interact, has an impact. Therefore, the higher the social vulnerability, the more vulnerable a society is to disasters, and it can be predicted that this will negatively affect livability. Cutter et al. [23] found that social vulnerability is defined by a high sensitivity to natural disasters and the degree of the ability to respond to disasters, and they used vulnerability, housing type, race, and sex as indicators. As for the studies related to the vulnerability assessment of ${\mathrm{PM}}_{2.5}$, many studies have investigated the effects of particulate matters on health [24,25]. The composition of the vulnerability index is mainly determined using response variables such as exposure, sensitivity, and adaptive capacity, according to the Intergovernmental Panel on Climate Change.

Most studies focused on exposure indices, such as the air pollution status and ${\mathrm{PM}}_{2.5}$ concentration, and did not consider potential risk factors, including ${\mathrm{PM}}_{2.5}$ vulnerability. Therefore, in this study, based on previous studies, social vulnerability to ${\mathrm{PM}}_{2.5}$ was considered to have a large effect on livability and was judged as an important indicator to be considered in the livability index. Figure 1 shows a conceptual diagram of the index system used for measuring livability considering ${\mathrm{PM}}_{2.5}$ vulnerability. Until now, few studies have empirically analyzed the relationship of ${\mathrm{PM}}_{2.5}$ with living-environment factors, such as livability, among geographically adjacent countries. As South Korea and China can geographically influence each other, it is necessary to consider how environmental pollutants such as ${\mathrm{PM}}_{2.5}$ affect the quality of the living environment, and how they can formulate a joint policy. Therefore, this study empirically analyzed how ${\mathrm{PM}}_{2.5}$ concentrations affect the livability index in geographically adjacent regions of South Korea and China.

First, we compare the correlation between ${\mathrm{PM}}_{2.5}$ concentration and the livability index in Korea and China. Second, we spatially analyze livability, considering the vulnerability of ${\mathrm{PM}}_{2.5}$. Third, to see the causal relationship between ${\mathrm{PM}}_{2.5}$ and the highly related livability index, panel regression analysis is performed to identify the effect. Based on the relationship between the ${\mathrm{PM}}_{2.5}$ concentration and the livability index, this study will contribute to a joint urban policy for ${\mathrm{PM}}_{2.5}$ reduction and response between the two countries in the future.

## 2. Materials and Methods

### 2.1. Study Area and Data

The study area included the Chinese eastern coastal region, where industrial complexes and large cities are concentrated, and entire regions of South Korea. China has three large cities (Beijing, Shanghai, and Tianjin), nine coastal regions (Liaoning, Hebei, Shandong, Jiangsu, Zhejiang, Fujian, Guangdong, Guangxi, and Hainan), and six central economic zones (Shaanxi, Henan, Anhui, Hubei, Hunan, and Jiangxi). South Korea has 34 regions, including seven large cities (Seoul, Busan, Incheon, Daegu, Daejeon, Gwangju, and Ulsan) and nine regions (Gyeonggi-do, Chungbuk, Chungnam, Jeonbuk, Jeonnam, Gangwon-do, Gyeongbuk, Gyeongnam, and Jeju-do). All these regions formed the study area (Figure 2). Annual data from 2015 to 2019 were used to analyze changes over time. For South Korea, ${\mathrm{PM}}_{2.5}$ data was available from 2015. To control for the impact of Covid-19, only data from 2015 to 2019 were used. For ${\mathrm{PM}}_{2.5}$ concentration data, IDW (Inverse Distance Weighted) spatial interpolation method was used to derive annual average values for each administrative district. The reason for using the IDW method among spatial interpolation methods was that the visualization and precision of the spatial distribution of PM2.5 concentration were superior to other interpolation methods [26] As shown in Table 1, data on variables that can be obtained from both South Korea and China were collected. For South Korea, annual data by province were collected from the national statistics portal (https://kosis.kr/index/index.do, accessed on 15 March 2022), and for China, data were collected from the national statistical yearbook [27], urban statistical yearbook [27], and environmental statistics yearbook [28].

### 2.2. Livability Measurement and Index

#### 2.2.1. Livability Measurement Methods

There are two methods to measure livability. The first estimation method is based on input and evaluates whether the living conditions are suitable for human needs and capacities. It uses a composite index of the economy (e.g., gross domestic product (GDP) per capita), society, environment, culture, education, etc. The second inductive evaluation method estimates health and satisfaction, life expectancy, and other outputs, such as happiness [29]. The former measures the objective status through statistical analysis of data sets, whereas the latter measures the satisfaction, perception, and happiness of residents.

Metrics and methods should be selected considering that livability is related to the quality of everyday social life and the quality of the environment with which people interact daily. The choice of indicators to include in the livability index often reflects a political agenda or is based on the subjectivity of researchers. Therefore, it is difficult to comprehensively measure livability because not all factors can be reflected [30]. Nevertheless, livability can objectively measure the quality of the living environment, which affects the quality of life. In this study, livability was measured based on the input estimation method, considering the effects of transboundary ${\mathrm{PM}}_{2.5}$. Furthermore, the results of this study were visualized and analyzed by the Geographic Information System (GIS) program in order to better understand and present the spatial–temporal variation of ${\mathrm{PM}}_{2.5}$ concentrations and livability levels in the study area. For spatial analysis, ArcGIS [31], a software developed by Esri, was used. Excel program was used for GRA analysis, and panel analysis was performed with STATA [32], a statistical program for time series regression analysis.

#### 2.2.2. Livability Index

We used the livability index reported by Jun et al. [33] (Table 1). The livability index used in this study was derived based on scientific and objective methodologies through a scoping review of previous studies. It considers eight domains: vulnerability, urban living infrastructure, urban planning, transportation, economy, society, safety, and environment. In previous studies, housing and satisfaction were also considered; however, in this study, they were excluded because of limitations in data acquisition. The livability index consists of 3 levels, 8 domains, 13 indicators, and 27 sub-indicators. The vulnerability index was composed of variables including vulnerable groups sensitive to exposure to ${\mathrm{PM}}_{2.5}$. For the vulnerable groups sensitive to ${\mathrm{PM}}_{2.5}$, the population under the age of 13, the population over 65, the aging index, and the number of beneficiaries of basic livelihood with high damage and low resilience in the event of a disaster and mortality were used as variables [1]. The reason for limiting this vulnerability to the vulnerable population index as a variable to measure vulnerability is that damage varies greatly depending on the population group with high disaster sensitivity and low response capacity [22].

### 2.3. Grey Relational Analysis (GRA)

GRA is a statistical method that analyzes multiple factors and is used when there is no clear relationship between the main influencing factors [34]. The GRA method reduces correlation errors caused by sample size limitations and trend uncertainty. Compared to commonly used mathematical statistical methods, such as analysis of variance, regression analysis, and principal component analysis, the GRA method has the advantage of producing consistent results for both quantitative and qualitative phenomena [35]. It can also be used to calculate the priority of influencing factors between ${\mathrm{PM}}_{2.5}$ concentration and the livability index. Livability indicators are composed of parameters in various domains that are influenced by ${\mathrm{PM}}_{2.5}$ concentrations. To study the relationship between ${\mathrm{PM}}_{2.5}$ concentrations and livability indicators, the variables of all indicators were standardized for different units of measurement. Indicators can be standardized based on the Z-score, the ratio of value, or min–max. The Z-score utilizes the mean and standard deviation to standardize a data point to a value between –1 and 1, the ratio of value method utilizes the maximum value to standardize a data point to a value between 0 and 1, and the min–max method normalizes data points to a value between 0 and 1 based on the maximum and minimum values. In this study, the min–max method was used. The ratio of value method has an advantage in that the value can be standardized as a positive number, but it is limited in that only the maximum value, not the range of the index is considered.

The theory of GRA is based on cybernetics and information theory, which can measure correlations, similarities, and differences between variables through mathematical calculations. By measuring differences between variables, it is possible to present a dynamic correlation between different subject characteristics [36,37]. The grey-related coefficient value calculated by the GRA method ranges between 0 and 1. The closer the coefficient value is to 0, the lower its relevance, and the closer to 1, the higher its relevance. In addition, the priority correlation between indicators can be displayed [38]. The GRA procedure is as follows:Suppose that the reference sequence and the sequences that are compared with the reference sequence after normalization are:
$$Y_{j}\left(t\right)=\left\{\begin{array}{cc} \begin{array}{cc} y_{1}\left(1\right) & y_{1}\left(2\right)\\ y_{2}\left(1\right) & y_{2}\left(2\right) \end{array} & \begin{array}{cc} \cdots & y_{1}\left(n\right)\\ \cdots & y_{2}\left(n\right) \end{array}\\ \begin{array}{cc} \cdots & \cdots \\ y_{i}\left(1\right) & y_{i}\left(2\right) \end{array} & \begin{array}{cc} \cdots & \cdots \\ \cdots & y_{i}\left(n\right) \end{array} \end{array}\right\}X_{ij}=\left\{\begin{array}{cc} \begin{array}{cc} x_{11} & x_{12}\\ x_{21} & x_{22} \end{array} & \begin{array}{cc} \cdots & x_{1j}\\ \cdots & x_{2j} \end{array}\\ \begin{array}{cc} \cdots & \cdots \\ x_{i1} & x_{i2} \end{array} & \begin{array}{cc} \cdots & \cdots \\ \cdots & x_{ij} \end{array} \end{array}\right\}$$

The reference column (*Y*) is the concentration of ${\mathrm{PM}}_{2.5}$. The comparator column (*X*) is a livability indicator.

2.Data normalization

Data standardization is necessary for the analysis and comparison of variables with different units. There are many data standardization methods, including initial value conversion, average value conversion, percentage conversion, Z-score conversion, and scale readjustment using maximum–minimum values [39,40,41]. This study used the min–max method for standardization. After normalization, all *Y_j_* values and *X_ij_* index values were normalized to values between 0 and 1. In data normalization, the division of expressions into positive data and negative data considers the negative impact on livability in the case of vulnerability and disaster indicators. For exponentiation, it was calculated considering the negative influence. In the case of ${\mathrm{PM}}_{2.5}$, concentration data was not standardized and was used as it is.
(1)$$Y_{j}=\frac{y_{i}-y_{i}min}{y_{i}max-y_{i}min}$$
(2)$$\mathrm{Positive}\;\mathrm{data}:\;X_{ij}=\frac{x_{ij}-x_{ij}min}{x_{ij}max-x_{ij}min}$$
(3)$$\mathrm{Negative}\;\mathrm{data}:\;X_{ij}=\frac{x_{ij}max-x_{ij}}{x_{ij}max-x_{ij}min}$$

3.Grey relational calculation

GRA determines whether data columns are closely related by comparing the similarity of the geometric shapes of the data column curves. The more similar these curves, the greater the association between the data. The grey relational calculation value after standardizing the data is obtained by Equations (4)–(6). In Equation (4), Δ*_ij_* is the difference between the dependent and independent variable columns, and *γ_ij_* is the relational coefficient. The value of the distinguishing coefficient (*ε*) in Equation (5) is generally set to 0.5. Equation (6) is used to calculate the grey correlation degree (*ρ_ij_*), and the order of the correlation degree can be expressed by the coefficient value. (4)$${∆}_{ij}=\left|y_{j}-x_{ij}\right|$$
(5)$$\gamma_{ij}=\frac{{∆}_{min}+\varepsilon {∆}_{max}}{{∆}_{ij}+\varepsilon {∆}_{max}}$$
(6)$$\rho_{ij}=\frac{1}{n}\sum_{i=1}^{n}\gamma_{ij}$$

4.The weight of the livability indicator

Livability was indexed by applying the R coefficient values calculated by GRA to the weighted values of the indicators. The weights were calculated using Equation (7), where $w_{i}$ represents the weight of index *i*.
(7)$$w_{i}=\frac{\rho_{ij}}{\sum_{t=2}^{n}\left|\rho_{ij}\left(t\right)\right|}$$

### 2.4. Panel Regression Analysis

Based on the GRA analysis results (Table 2), the livability indicators with ${\mathrm{PM}}_{2.5}$ concentrations and grey correlation degree higher than 0.6 were selected, and panel data regression analysis was performed. Before analyzing the panel data regression model, a multicollinearity test was performed on each panel data using the Variance Inflation Factor (VIF) index. VIF is the coefficient of variance expansion, and the VIF value is the most commonly used method for multicollinearity testing [42,43,44,45]. According to the VIF method, if the VIF value of the root variable is less than 10, it proves that there is no multicollinearity problem in the model [43,44,46]. Accordingly, in this study, variables with VIF values greater than 10 were removed from the model. In addition, before estimating the panel data regression model, the hypothesis compliance of the model was always tested. In particular, time series autocorrelation and heterogeneity tests were performed for the characteristics of the panel data [47]. Moreover, in this study, the heteroscedasticity and autocorrelation of the population were tested using the modified Wald test and the Wooldridge test, respectively. Panel regression models usually test the significance of fixed and random effects on individual and time effects, and panel models combining cross-sectional and time series data are used for research purposes. If both effects are significant, the Hausman test is used to set the characteristics of the data and the appropriate model as the final analytical model [47].

In addition, this study performs a natural logarithmic transformation on the data. The natural logarithm transformation can reduce the heterogeneity of the data without changing the original characteristics of the data. In addition, the inconvenience of different units of measurement can be eliminated, and the estimated coefficients are elastic moduli, with the advantage that a 1% increase in the independent variable can be interpreted as a net percentage change in the dependent variable [48]. Therefore, the regression analysis Equation (8) for panel data in this study is as follows:(8)$$\;{\mathrm{PM}}_{2.5}{}_{it}=\alpha_{i}+\beta_{1}Vulnerability_{it}+\beta_{2}Infrastructure_{it}+\beta_{3}Plan_{it}\;\;\;\;\;\;\;+\beta_{4}Transportation_{it}+\beta_{5}Economic_{it}+\beta_{6}Social_{it}\;\;\;\;+\beta_{7}Safety_{it}+\beta_{8}Environment_{it}+\mu_{it}$$

## 3. Results

### 3.1. Grey Correlation between
${\mathrm{PM}}_{2.5}$ Concentration and the Livability Indicators

Table 2 shows the results of the grey correlation degree between ${\mathrm{PM}}_{2.5}$ concentration and the livability indicators. The correlation between factors is considered high when the grey correlation degree is greater than 0.6 [37]. The grey correlation degree of ${\mathrm{PM}}_{2.5}$ concentration to the livability indicators in South Korea ranged from (0.459) to (0.870). In the livability index, safety aspects had the highest correlation with ${\mathrm{PM}}_{2.5}$ concentration, especially the traffic safety index of traffic accidents per 1000 vehicles (0.890). Existing research has related traffic indicators to the emission of air pollutants such as particulate matters [10]. The urban living infrastructure (0.665) was the second most relevant indicator, and the culture and leisure (0.825) and education (0.687) indicators were highly correlated with ${\mathrm{PM}}_{2.5}$ concentration. As for the sensitivity indicator for vulnerability, the highest correlation was found for the percentage of the population under 13 years of age (0.870). Regarding urban living infrastructure, high correlations were observed for the education sub-indicator of the number of students per teacher (0.687) and the culture and leisure sub-indicator of the urban park area per 1000 population (0.826). The public administration budget in general accounting (0.746) in the social inclusion indicator in the social development domain, the number of traffic accidents per 1000 vehicles (0.840) in the traffic safety indicator in the safety domain, and the annual average precipitation (0.677) and annual average temperature (0.640) in the climate indicator in the environment domain, were highly correlated with ${\mathrm{PM}}_{2.5}$ concentration.

The results of the livability correlation coefficient calculation for China showed that urban living infrastructure (0.773), urban planning (0.737), and vulnerability (0.673) were highly correlated with ${\mathrm{PM}}_{2.5}$ concentration, with the highest correlation coefficient for urban living infrastructure. Specifically, the health and healthcare (0.932) indicator in the urban living infrastructure area showed the highest correlation, and the urban park area per 1000 population (0.856) in the culture and leisure indicator also showed a high correlation with ${\mathrm{PM}}_{2.5}$ concentration. Unlike that in South Korea, the correlation of urban planning was high in China (0.737), especially that of the green area rate (0.890), which was very high. In the domain of vulnerability, sensitivity indicators, such as the ratio of recipients basic living support (0.874) and the aging index (0.931), showed very high correlations. The livability indicator that was highly correlated with ${\mathrm{PM}}_{2.5}$ concentration in both South Korea and China was urban living infrastructure. The number of urban parks, final energy consumption, and the average annual temperature had a high relevance in both two countries.

This is consistent with findings in previous studies. Traffic safety is related to the rate of automobile accidents. Traffic is a major contributor to ${\mathrm{PM}}_{2.5}$ as indicated by empirical studies [49,50]. Urban parks, a culture and leisure sub-indicator, are green spaces that have the effect of reducing ${\mathrm{PM}}_{2.5}$ concentration [51,52,53]. In this study, the coefficient of correlation between the urban park area and ${\mathrm{PM}}_{2.5}$ concentration was above 0.8, which is very high, in both South Korea and China. Energy consumption is closely related to ${\mathrm{PM}}_{2.5}$ concentration and is used as a carbon footprint indicator [34]. In South Korea, the average annual temperature and precipitation, which are meteorological factors, were more strongly correlated with ${\mathrm{PM}}_{2.5}$ concentration than in China. Meteorological factors such as temperature, humidity, wind speed, and precipitation are important parameters affecting particulate matter retention [54]. In South Korea, the education index showed a characteristically high correlation with ${\mathrm{PM}}_{2.5}$ concentration, which is explained by the high urbanization rate and the population density centered in educational areas. In China, urban living infrastructure and urban planning were highly correlated, which is in line with the results of previous studies showing that the urbanization level index is highly related to ${\mathrm{PM}}_{2.5}$ in China [55]. Finally, in China, socio-economically disadvantaged groups are more vulnerable to ${\mathrm{PM}}_{2.5}$ than in South Korea.

### 3.2. Spatial Analysis of the Livability Index of South Korea and China

The livability index was calculated based on the livability indicators and relevance coefficients derived from GRA as weights. Spatial analysis of the livability index was performed using the GIS program. Annual average ${\mathrm{PM}}_{2.5}$ concentrations were divided into 15 classes using the natural breaks classification method, and the results are shown in (Figure 3). The natural breaks approach minimizes the average deviation of the overall values within the same class of data and maximizes the dispersion between classes [56]. According to the WHO, the overall ${\mathrm{PM}}_{2.5}$ concentration in South Korea is lower than that in China. In both countries, ${\mathrm{PM}}_{2.5}$ concentrations decreased from 2015 to 2019. For South Korea, ${\mathrm{PM}}_{2.5}$ concentration was the highest in the Jeon-buk region in 2015 and has declined since 2018. Gyeonggi-do and Gangwon-do also show decreasing ${\mathrm{PM}}_{2.5}$ concentration trends as of 2018. The highest ${\mathrm{PM}}_{2.5}$ concentrations were noted in Gyeonggi-do, in the northern Chungcheong Province, and in Jeon-nam, in the west coast region close to China. Gyeonggi-do is a densely populated region near Seoul, and Chung-buk also is a region with high population mobility. Moreover, the distribution of industrial facilities is concentrated in the Gyeonggi-do and Chung-buk regions. In the Chung-buk region, ${\mathrm{PM}}_{2.5}$ concentrations decreased in 2016 but have been increasing since 2017 and are currently the highest in the country. A recent study of the inflow path of fine dust in the Chung-buk area showed foreign inflow from China and domestic fine dust generation by industrial facilities and thermal power plants located in Chung-buk and surrounding coastal areas [57].

Regarding the spatial distribution of the annual average ${\mathrm{PM}}_{2.5}$ concentrations from 2015 to 2019, except in the Fujian, Guangdong, and Hainan regions, ${\mathrm{PM}}_{2.5}$ concentrations were generally relatively high in 2015. They were the highest in the central and eastern regions, where primary industries, such as the coal industry, are concentrated. It is well known that especially the secondary sector contributes to air pollution [58,59]. While ${\mathrm{PM}}_{2.5}$ concentrations in China have been gradually decreasing between 2015 and 2019, those in the central and eastern regions still exceed the WHO air quality norms. The regions with the highest ${\mathrm{PM}}_{2.5}$ concentrations in 2019 were mainly the Jing-Jin-Ji (Beijing-Tianjin-Hebei) area and surrounding areas (Shandong and He-nan). These regions have high urbanization rates and GDP per capita.

Figure 4 shows the livability measurement results considering ${\mathrm{PM}}_{2.5}$ vulnerability. China showed a higher vulnerability to ${\mathrm{PM}}_{2.5}$ and lower livability than South Korea. From 2015 to 2019, livability improved in both South Korea and China. In South Korea, the Gangwon-do and Gyeong-buk regions showed the highest livability in 2015, and in 2017, Ulsan and Daegu showed the highest livability among the Gyeonggi-do, Chung-nam, and Jeonbuk regions and large cities. In 2019, Incheon and Jeon-buk showed the highest livability among large cities, followed by Jeon-nam, Chung-buk, and Gangwon-do. On Jeju Island, livability has been declining since 2018. In Gangwon-do, regional development projects for balanced regional development are being actively carried out. Jeon-nam also showed a high livability; the urban living infrastructure index and planning index were higher in this region than in other areas. In the coastal region of southern China, ${\mathrm{PM}}_{2.5}$ concentrations and vulnerability were low. Most of the areas with higher livability had a lower ${\mathrm{PM}}_{2.5}$ vulnerability index. The livability index of coastal areas was higher than that of inland areas. Vulnerability indices were lower in major cities, such as Beijing and Tianjin, and higher in surrounding areas (Tianjin and Hebei). This suggests a concentration of vulnerable populations in the surrounding areas. In inland China, livability has been declining since 2016.

### 3.3. Panel Regression Analysis Results

Table 3 shows the results of panel regression analysis by deriving an indicator with a grey correlation coefficient of 0.6 or higher. In the case of South Korea, the random effects model was significant in the Hausmann test. The feasible generalized least squares (FGLS) regression model that can correct for autocorrelation and heteroscedasticity was the most suitable, and the results are interpreted using this model. In South Korea, it was found that education, health and medical care, social inclusion, traffic safety, and climate indicators in the urban living infrastructure domain affect ${\mathrm{PM}}_{2.5}$ concentration. Educational indicators in the urban living infrastructure domain were significant in the positive direction, meaning that the better the educational indicators, the lower the ${\mathrm{PM}}_{2.5}$ concentration. The higher the quality of education, the more abundant human capital, which increases economic productivity, which in turn improves livability [12,60,61]. In other words, an area with good livability means that the quality of the environment is high, so it can be interpreted that ${\mathrm{PM}}_{2.5}$ concentration is low. The safety indicator was also significant in the positive direction, and the number of traffic accidents is likely to be high in a place with a lot of vehicle movement, so ${\mathrm{PM}}_{2.5}$ concentration is likely to increase. The social inclusion indicator in the social domain was also shown to be significantly related to ${\mathrm{PM}}_{2.5}$ concentration. The revenue of the local general accounting and administrative budget is mainly tax revenue. Accordingly, an increase in tax revenue means an increase in urban growth or development activities [62].

In the case of China, the fixed-effects model was found to be significant in the Hausmann test, and the results were interpreted based on the Driscoll–Kraay model, which can correct for autocorrelation and heteroscedasticity. Most of the indicators with high GRA relevance were similar to those in South Korea. As a result of the panel regression analysis, the cultural, leisure, education, and health and medical care indicators in the urban living infrastructure domain were found to be significant, and the economic vitality, social inclusion, and climate indicators were also found to have an effect on ${\mathrm{PM}}_{2.5}$ concentration. Educational indicators were found to be influenced by wealth, in common with South Korea, and economic indicators showed that ${\mathrm{PM}}_{2.5}$ concentration increased as the number of workers increased. Health and medical care indicators were different from those in South Korea, with a positive effect on ${\mathrm{PM}}_{2.5}$ in China based on the influence of wealth. Health and medical care indicators refer to access to medical services, and in general, the higher the number, the higher the livability. However, in the case of China, the greater the access to medical services, the higher was the ${\mathrm{PM}}_{2.5}$ concentration. However, these specific factors cannot be identified in this study. In both South Korea and China, the vulnerability indicators, which showed a high correlation in the GRA, did not appear significant in the panel regression model.

## 4. Discussion

The environment is a major determinant of livability, and air quality in particular is known to have the greatest impact [11,12]. ${\mathrm{PM}}_{2.5}$ travels long distances into the atmosphere and affects neighboring countries, so it is important to understand the influence relationship between neighboring countries [6,7]. The livability index is a standard for measuring the quality of the urban environment and the basis for protecting citizens from environmental pollution and the right to live in a safe environment [9]. Therefore, in this study, the effect of ${\mathrm{PM}}_{2.5}$ concentration on livability in neighboring regions of China and South Korea was investigated. As an alternative to transboundary ${\mathrm{PM}}_{2.5}$, international cooperation between South Korea and China has been carried out. Recognizing that ${\mathrm{PM}}_{2.5}$ measures are a priority, the two countries agreed to exchange information on ${\mathrm{PM}}_{2.5}$ component analysis. The South Korea and China Air Quality Joint Research Center conducts joint research in 10 areas, including the identification of the causes of air pollution, improvement of forecasting models, and reduction of air pollutant emissions. It was agreed to establish a research plan. However, this has limitations in responding because it is closely linked to the difference in the mutual contribution rate of air pollution between the two countries and to political and economic issues.

As confirmed in previous studies, ${\mathrm{PM}}_{2.5}$ is not limited to environmental problems and affects daily life. It is necessary to identify and respond at the local level because it is influenced by various factors such as climate, geography, economy, society, and population of the region. Through the GRA analysis, it was confirmed that the urban life infrastructure area showed the highest correlation between South Korea and China.

In China, urban living infrastructure, urban planning, and vulnerability were found to be highly correlated with the ${\mathrm{PM}}_{2.5}$ concentration. Urban living infrastructure, which measures the services and facilities required to live, including educational opportunities, access to healthcare, cultural and recreational facilities, public amenities, and parks, is the major factor in the livability index. The availability and accessibility of urban living infrastructure are also related to urban planning and transportation. When livability was defined based on the quality of life, the quality of the external environment, and living conditions, urban living infrastructure, planning, and transportation had the highest weights among all indicators [63]. The high correlation between urban living infrastructure and ${\mathrm{PM}}_{2.5}$ concentration indicates that the latter clearly affects the quality of the living environment. Conversely, this means that the indicators that make up the livability index can also influence ${\mathrm{PM}}_{2.5}$ concentrations. Through the panel regression analysis, the influence relationship could be identified more clearly. In both South Korea and China, the indicators of urban living infrastructure, society, and environment were significant. Unlike China, in South Korea, traffic safety indicators were significant, and in China, the area of urban parks per 1000 people and the number of businesspeople were significant.

This study revealed that ${\mathrm{PM}}_{2.5}$ affects our daily life and is not just an environmental problem. Based on our findings, we suggest several measures that can be taken to tackle the problem of ${\mathrm{PM}}_{2.5}$ pollution. First, in the urban living infrastructure domain, urban parks and green spaces were highly negatively correlated with ${\mathrm{PM}}_{2.5}$ concentration, and they play a role in reducing ${\mathrm{PM}}_{2.5}$ concentrations. Thus, to reduce the negative impact of ${\mathrm{PM}}_{2.5}$ in cities, it is necessary to increase the ratio of urban parks or green spaces. In addition, in China, ${\mathrm{PM}}_{2.5}$ concentration showed a very high correlation with the health and medical care index. Indeed, in China, ${\mathrm{PM}}_{2.5}$-related deaths increased by 23% between 2002 and 2017 [64]. Second, although there was not a clear causal relationship between ${\mathrm{PM}}_{2.5}$ concentration and social vulnerability, the GRA analysis showed a high correlation. In South Korea, the population under 13 years of age showed a high correlation, whereas, in China, the ratio of recipients of basic living support and the elderly population showed high correlations. High-vulnerability areas have potential risks and therefore have the potential to have lower livability in the long run. In both South Korea and China, areas surrounding cities and primary or secondary industrial centers were more vulnerable than large cities. China has a high ${\mathrm{PM}}_{2.5}$-related mortality rate and high vulnerability. Thus, more proactive countermeasures are needed. Third, in the environmental domain, energy consumption showed a high correlation with ${\mathrm{PM}}_{2.5}$ concentration in both South Korea and China. Energy consumption is directly related to ${\mathrm{PM}}_{2.5}$ concentration as it has a direct environmental impact. Finally, South Korea and China are geographically close to each other and can affect each other in terms of pollution. According to a South Korea–China–Japan joint research report, 32% of the total ${\mathrm{PM}}_{2.5}$ concentration in South Korea was transboundary ${\mathrm{PM}}_{2.5}$ from China, and the ${\mathrm{PM}}_{2.5}$ contribution rate of South Korea to China was 2% [65]. Therefore, for South Korea, control of the ${\mathrm{PM}}_{2.5}$ pollution in China is of utmost importance. In addition, based on the correlations among the livability indicators, the neighboring regions of China and South Korea need to develop more specific and practical countermeasures for urban living infrastructure, urban planning, transportation, and vulnerability factors that are highly correlated with ${\mathrm{PM}}_{2.5}$ not only at the national level, but also at the regional level.

As a limitation of this study, when collecting the data for measuring livability, only variables that were in common between the two countries were extracted. Therefore, various other potentially relevant indicators, such as the housing index, were not included. In future, it is necessary to lay a foundation for establishing regional ${\mathrm{PM}}_{2.5}$ reduction measures by a South Korea–China joint council based on studies.

## 5. Conclusions 

The purpose of this study was to investigate the effect of the concentration of ${\mathrm{PM}}_{2.5}$ on the livability of neighboring regions of China and South Korea with the aim of informing common policy development. South Korea and China have shared an analysis of ${\mathrm{PM}}_{2.5}$ emissions and movement through joint research and an agreement to reduce ${\mathrm{PM}}_{2.5}$. However, there is a limit to the response due to the difference in the mutual contribution rate of ${\mathrm{PM}}_{2.5}$ emission between the two countries, and political and economic issues. Therefore, this study aims to contribute to the response at the regional level by empirically analyzing the effect of ${\mathrm{PM}}_{2.5}$ concentration on the livability index of South Korea and China.

As a result, urban living infrastructure showed the highest correlation and influence with ${\mathrm{PM}}_{2.5}$ concentration in both countries. Urban living infrastructure occupies the largest portion of the livability index and includes all physical environments and services enjoyed in daily life in the city. Urban living infrastructure is closely related to planning and policy indicators because it is implemented as a plan and policy for environmental quality, appropriate distribution, and accessibility [63]. In addition, as a result of spatial analysis to examine changes in ${\mathrm{PM}}_{2.5}$ concentration and livability according to time and space, the average annual ${\mathrm{PM}}_{2.5}$ concentration decreased from 2015 to 2019 in both South Korea and China, and livability improved. At the regional level, ${\mathrm{PM}}_{2.5}$ concentrations were high in areas surrounding large cities, and livability was low. In China, the vulnerability index was also high in the areas around big cities, and the livability in the inland regions tended to be lower. Overall, South Korea showed a lower ${\mathrm{PM}}_{2.5}$ concentration and higher livability than China. The concentration of ${\mathrm{PM}}_{2.5}$ was highest in areas with high population density and industrial complexes. Since the spatial distribution of ${\mathrm{PM}}_{2.5}$ concentration and the livability index differ according to regions, a regional approach is necessary for a joint response to reduce ${\mathrm{PM}}_{2.5}$ at the national level. Therefore, it is necessary to prepare policies at the regional level to respond to ${\mathrm{PM}}_{2.5}$ between bordering countries.

In this study, regional countermeasures can be prepared according to the livability index that is greatly affected by ${\mathrm{PM}}_{2.5}$. In the case of urban living infrastructure, accessibility and population density are major planning factors and are related to land use and urban form. In the case of China, efforts are needed to reduce the concentration of ${\mathrm{PM}}_{2.5}$ by expanding urban parks and green areas. In particular, in inland areas, the vulnerability is increasing, so it seems urgent to prepare policies for the vulnerable population. South Korea needs an active response to ${\mathrm{PM}}_{2.5}$ reduction through urban planning to reduce the use of automobiles and link it with transportation policies. In areas where industrial complexes are concentrated, the concentration of ${\mathrm{PM}}_{2.5}$ is high, so it seems that separate countermeasures are needed. Lastly, environmental indicators have also been shown to have a significant impact, and complementary policies can be prepared by linking them with climate change response policies and ${\mathrm{PM}}_{2.5}$ reduction measures. Through empirical analysis, this study revealed that the decrease in ${\mathrm{PM}}_{2.5}$ concentration is not an environmental problem but is closely related to the quality of the environment we can experience in our daily life. In addition, this study is meaningful as a basic study for preparing policies to respond not only at the national level, but also at the regional and city level in preparing a joint response aimed at reducing ${\mathrm{PM}}_{2.5}$ between South Korea and China in the future.

## Figures and Tables

**Figure 1 ijerph-19-13049-f001:**
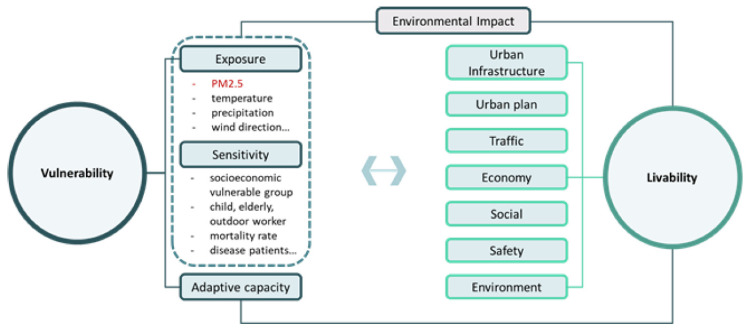
Conceptual diagram of livability measurement considering PM_2.5_ vulnerability.

**Figure 2 ijerph-19-13049-f002:**
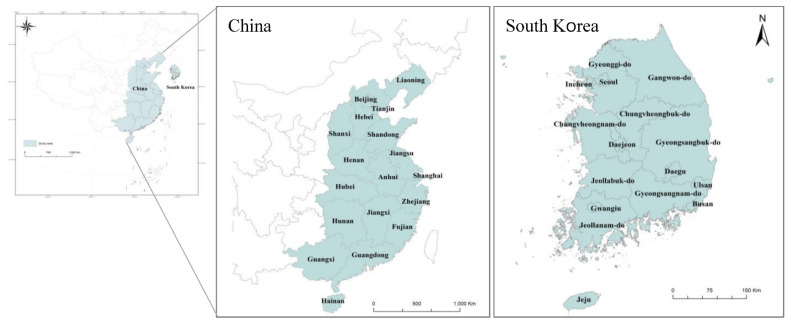
Study area.

**Figure 3 ijerph-19-13049-f003:**
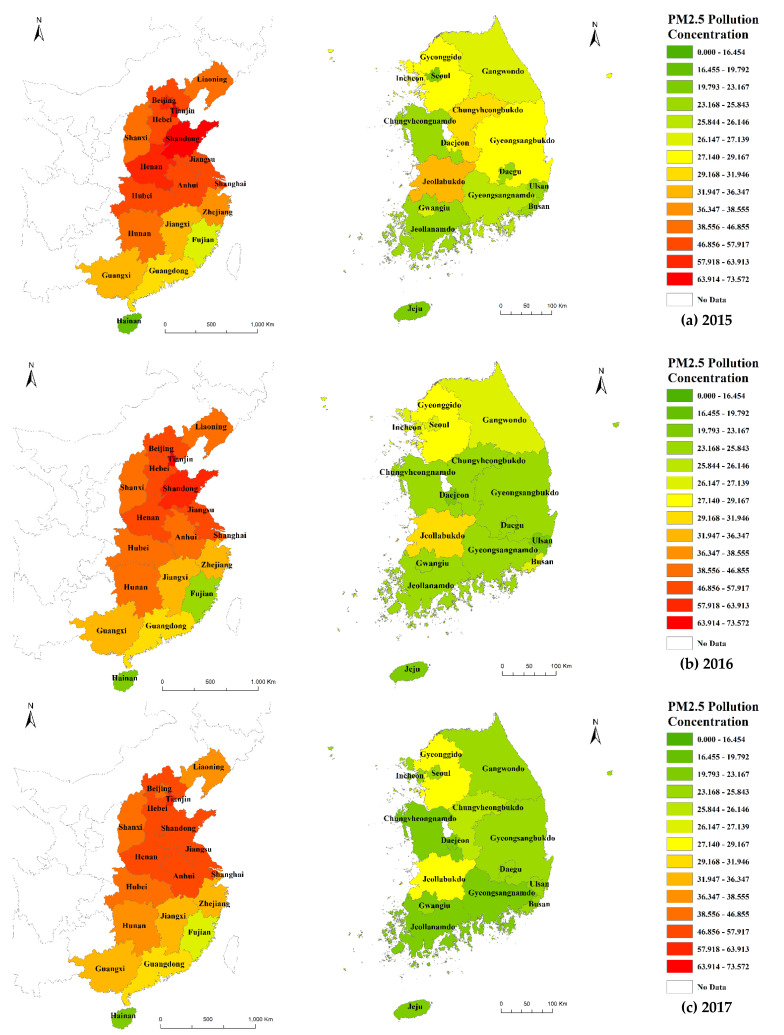
${\mathrm{PM}}_{2.5}$ concentrations.

**Figure 4 ijerph-19-13049-f004:**
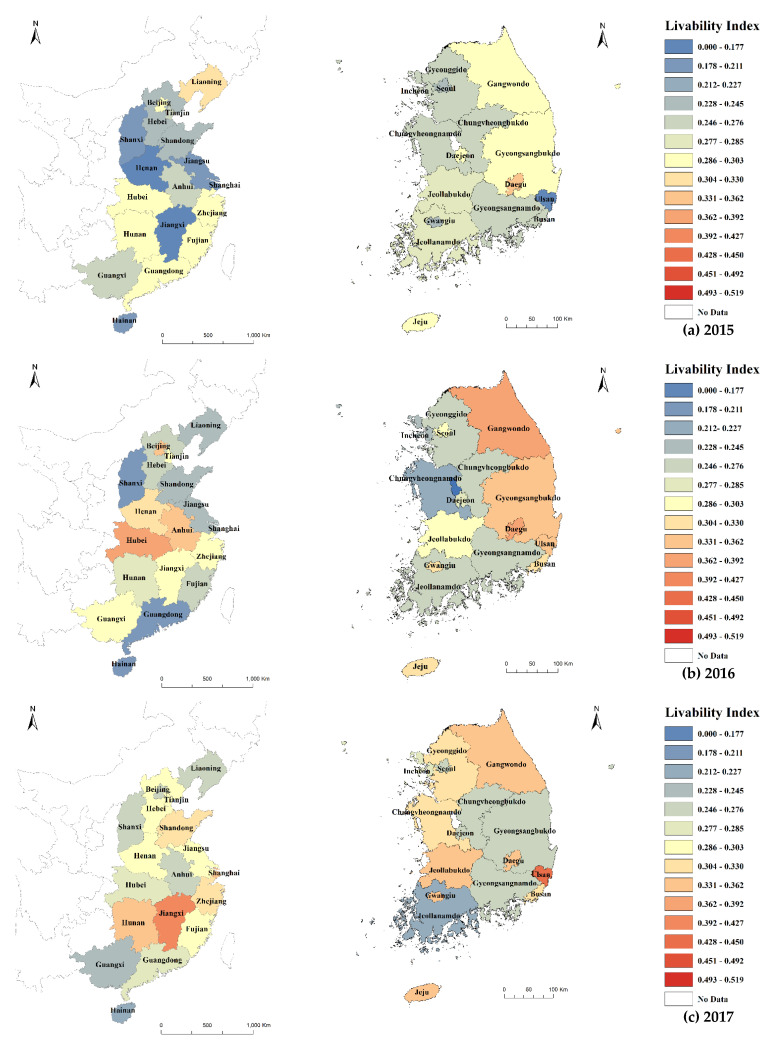
Livability measurement results considering ${\mathrm{PM}}_{2.5}$ vulnerability.

**Table 1 ijerph-19-13049-t001:** Livability indicators used in this study.

Domain	Indicators	Sub-Indicators
Vulnerability	Sensitivity	X1: Percentage of the population over 65 years X2: Percentage of the population under 13 years X3: Ratio of recipients of basic living support X4: Mortality rate per 100,000 population X5: Aging index
Urban living infrastructure	Culture and leisure	X6: Urban Park area per 1000 population
	Education	X7: Number of students per teacher X8: Number of childcare facilities per 1000 children
	Health and healthcare	X9: Number of employees in medical institutions per 1000 population X10: Hospital beds per 1000 population
Urban plan	Urban growth	X11: Urban area per capita X12: Green area rate
Transportation	Convenience of movement	X13: Road pavement rate X14: Number of vehicle registrations per person
Economic development	Economic vitality	X15: GDP per capita X16: Income per capita X17: Economic participation rate X18: Number of employees per 1000 population
Social development	Social inclusion	X19: Public administration budget in general accounting X20: Social welfare budget in general accounting X21: Population
Safety	Traffic safety	X22: Number of traffic accidents per 1000 vehicles
	Natural disaster	X23: Natural disaster damage
Environment	Environmental consumption	X24: Final energy consumption
	Water management system	X25: Water supply rate
	Climate	X26: Annual average precipitation X27: Annual average temperature

**Table 2 ijerph-19-13049-t002:** Grey correlation degree results for the livability indicators.

Domain	Grey Correlation Degree		Indicators	Grey Correlation Degree		Sub Indicators	Grey Correlation Degree	
	Korea	China		Korea	China		Korea	China
Vulnerability	0.565	**0.673**	Sensitivity	0.565	**0.673**	X1: Percentage of the population over 65 years	0.497	0.530
						X2: Percentage of the population under 13	**0.870**	0.512
						X3: Ratio of recipients of basic living support	0.470	**0.874**
						X4: Mortality rate per 100,000 population	0.493	0.518
						X5: Aging index	0.496	**0.931**
Urban living infrastructure	**0.665**	**0.773**	Culture and leisure	**0.825**	**0.856**	X6: Urban park area per 1000 population	**0.825**	**0.856**
			Education	**0.687**	0.561	X7: Number of students per teacher	**0.870**	0.512
						X8: Number of childcare facilities per 1000 children	0.504	**0.610**
			Health and healthcare	0.483	**0.932**	X9: Number of employees in medical institutions per 1000	0.481	**0.934**
						X10: Hospital beds per 1000 population	0.485	**0.931**
Urban plan	0.504	**0.737**	Urban growth	0.504	**0.737**	X11: Urban area per capita	0.526	0.584
						X12: Green area rate	0.482	**0.890**
Transportation	0.505	0.523	Convenience of movement	0.505	0.523	X13: Road pavement rate	0.483	0.518
						X14: Number of vehicle registrations per person	0.527	0.527
Economic development	0.495	0.590	Economic vitality	0.495	0.590	X15: GDP per capita	0.504	0.474
						X16: Income per capita	0.483	0.526
						X17: Economic participation rate	0.461	0.514
						X18: Number of employees per 1000 population	0.531	**0.844**
Social development	0.556	0.475	Social inclusion	0.556	0.475	X19: Public administration budget in general accounting	**0.746**	0.467
						X20: Social welfare budget in general accounting	0.462	0.452
						X21: Population	0.460	0.507
Safety	**0.672**	0.502	Traffic safety	**0.840**	0.429	X22: Number of traffic accidents per 1000 vehicles	**0.840**	0.429
			Natural disaster	0.504	0.574	X23: Natural disaster damage	0.504	0.574
Environment	0.580	0.541	Environmental consumption	**0.624**	0.512	X24: Final energy consumption	**0.624**	0.512
			Water management	0.459	0.475	X25: Water supply rate	0.459	0.475
			Climate	**0.658**	0.540	X26: Annual average precipitation	**0.677**	0.462
						X27: Annual average temperature	**0.640**	**0.616**

Note: The distinguishing coefficient was valued at 0.5. Values with a correlation coefficient of 0.6 or more were bolded.

**Table 3 ijerph-19-13049-t003:** Panel regression analysis results.

Domain	Indicators	Sub Indicators	South Korea				China			
			**(1)**	**(2)**	**(3)**	**(4)**	**(5)**	**(6)**	**(7)**	**(8)**
Vulnerability	Sensitivity	X3: Ratio of recipients of basic living support	0.013	0.099	0.053	0.001				
Urban living infrastructure	Culture and leisure	X6: Urban park area per 1000 population	0.004	−0.170 *	−0.085	0.004	−0.142	0.080	−0.049	**0.080 ***
	Education	X7: Number of students per teacher	0.587 ***	1.819 ***	0.787 ***	**0.627 *****				
		X8: Number of childcare facilities per 1000 children	−0.146	−0.267	−0.158	**−0.158 ***	−0.161	−0.478 **	−0.182	**−0.478 *****
	Health and healthcare	X9: Number of employees in medical institutions per 1000	−0.346 ***	0.644	−0.358 *	**−0.335 *****	0.164	0.8317	0.273	**0.832 *****
		X10: Hospital beds per 1000 population	0.104 *	−0.047	0.156	**0.127 *****	0.515 *	0.197	0.761 **	**0.197 *****
Urban plan	Urban growth	X12: Green area rate	−0.118	0.013	−0.015	−0.121	0.047	−0.044	−0.499	−0.044
Economic development	Economic vitality	X18: Number of employees per 1000 population	−0.086	0.462	0.061	−0.034	0.637	1.992 *	0.980 *	**1.992 ***
Social development	Social inclusion	X19: Public administration budget in general accounting	0.081	0.225 **	0.113	**0.107 *****	−0.344	−0.821 **	−0.457**	**−0.821 *****
Safety	Traffic safety	X22: Number of traffic accidents per 1000 vehicles	0.210 ***	0.063	0.116	**0.191 *****	0.083	−0.057	−0.001	−0.057
Environment	Environmental consumption	X24: Final energy consumption	0.228	−0.057	−0.046	0.045	0.057	0.142	0.135	0.142
	Climate	X26: Annual average precipitation	−0.061 *	−0.054	−0.075 **	**−0.059 *****	−0.354 ***	−0.197	−0.241 ***	**−0.197 *****
		X27: Annual average temperature	−0.315 **	−0.029	−0.264	**−0.442 *****				
		_cons	3.014	−1.579	2.292	3.252 ***	1.064	−7.163	−0.476	−7.163
R-squared			0.597	0.679	0.630		0.606	0.522	0.461	0.522
Modified Wald test		chi2 (16)	1268.80				2010.10			
		Prob > chi2	0.000				0.000			
Wooldridge test		F(1, 15)	16.450				10.456			
		Prob > F	0.001				0.005			
Hausman test		chi2 (13) = (b-B)’[(V_b-V_B)^(−1)](b-B)	12.320				202.410			
		Prob > chi2	0.501				0.000			

Note 1: Significant levels, * *p* < 0.1; ** *p* < 0.05; *** *p* < 0.01. Note 2: Model (1): Pooled OLS; Model; Model (2): Fixed-effect Model; (3): Random-effect Model; (4): Feasible Generalized Least Squares; Model (5): Pooled OLS; Model; Model (6): Fixed-effect Model; (7): Random-effect Model; Model (8): Fixed-effect regression with Driscoll–Kraay standard errors. Note 3: Statistically significant numbers were expressed in bold.

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
