# Peer review of "Air Pollution (PM2.5) Negatively Affects Urban Livability in South Korea and China"

_ijerph, 2022, doi:10.3390/ijerph192013049_

Round 1

Reviewer 1 Report (Previous Reviewer 1)

I had reviewed this manuscript before and provided detailed comments/suggestions. This resubmitted version has been improved though not all the comments are satisfactorily addressed. An example of the unaddressed comment is as follows:

How did the authors interpolate the PM2.5 data over the study area? What is the accuracy of interpolated concentrations? Did the authors validate their results?

I suggest that authors address all previous comments properly while revising the manuscript. 

Author Response

We are happy to respond to your comments and submit a revised manuscript. We appreciate your suggestions and comments. The corrections were reflected in revised version. The part modified by the comment was marked in red. Comments and answers are indicated by inserting memo into the manuscript. Once again, thank you very much for reviewing the manuscript and providing advice.

The response to your comments can be found in the attachment.

Reviewer 2 Report (New Reviewer)

Dear authors,

I understand you are studying correlationship between PM2.5 and urban livability in South Korea and China, to emphasize transboundering pollutant transport. Then, why in discussion section (L9), you speak about Japan?

In page #4, there should be figure #2, I cannot see it. I suppose, in that figure readers can view where is South Korea in respect to China, and what is the main atmospheric circulation (per month? season? year?). Otherwise, figures #3 and #4 do not show transboundering issues at all.

As you say in section #2.3 GRA is based on cybernetics and information theory. It was not until 2013 that   Pai, T.-Y., Hanaki, K. and Chiou, R.-J. (2013), Forecasting Hourly Roadside Particulate Matter in Taipei County of Taiwan Based on First-Order and One-Variable Grey Model. Clean Soil Air Water, 41: 737-742. 

proved that Grey theory was usable for pollutants study. I think you should know it.

Your data are really annual? Do you mean you are working with annual pollution data? How do you calculate it? So, you really have one value for each city?

In page #6 you say you normalize your data. You have one equation (2) for positive data, and one equation (3) for negative data.

How can you have negative PM2.5 concentration? 

On top of this, concentrations do not belong to R but to R+. In that space, sum is not a closed operation. So it is absolutely wrong to sum or subtract concentration values. 

They are percentages, so they are normalized. Why do you do it so complicated?

Table #4 shows livability indicators used in this study. Where is the equation to calculate livability index? You say "livability index was derived based on scientific...", but, how do you calculate it?

Then it is an index, so, it does not have units. Again, why do you try to normalize something whose value is between 0 and 1?

In page #5 you say that Y variable states for PM2.5 concentration. In page #7 you define Y as a regression analysis. Please, could you use the same letters for the same variables? 

It is really poor to have conclusions on just 4 data per city. 

Author Response

We are happy to respond to your comments and submit a revised manuscript. We appreciate your suggestions and comments. The corrections were reflected in revised version. The part modified by the comment was marked in red. Comments and answers are indicated by inserting memo into the manuscript. Once again, thank you very much for reviewing the manuscript and providing advice.

The response to your comments can be found in the attachment.

Round 2

Reviewer 1 Report (Previous Reviewer 1)

The manuscript has been improved. Please shorten the "conclusion" section (e.g., first paragraph of the conclusion seems unnecessary). Only the key findings and implications of the study can be presented concisely in the conclusion section. 

Author Response

We are happy to respond to your comments and submit a revised manuscript. We appreciate your suggestions and comments. The corrections were reflected in revised version. The part modified by the comment was marked in red. Comments and answers are indicated by inserting memo into the manuscript. Once again, we sincerely thank you for reviewing and giving advice on our article.

Thank you for your suggestion and pointing out the problems. Blue text is a response to a comment. Details are marked in red with notes in the manuscript.

1.   The manuscript has been improved. Please shorten the "conclusion" section (e.g., first paragraph of the conclusion seems unnecessary). Only the key findings and implications of the study can be presented concisely in the conclusion section. 

-> (p18, 2-43) The contents of the conclusion part have been shortened and rewritten.

We sincerely thank you for reviewing and giving advice on our article.

We've improved a lot compared to the beginning by diligently modifying and supplementing as you advised. Thank you very much.

Reviewer 2 Report (New Reviewer)

Dear authors, 

you have improved your paper. However, it does not have statistical soundness: using just one data per year and town is not statistically valid.

In your paper still appears y to indicate different variables. That makes a confusion to the reader.

In my comment about normalization, you replied:

'In data normalization, the division of expressions into positive data and negative data considers the negative impact on livability in the case of vulnerability and disaster indicators. For exponentiation, it was calculated considering the negative influence.

PM2.5 concentration data was not standardized and was used as it is.'

But this is not explained in the text. So readers can be confused (eq. 1, 2 3 in lines 219-221)

In your conclusions, you say you have proved L#8 'according to the geographical and social characteristics of the country'  Well, I have not been able to see any geographic map, with mountains, rivers.... How does the air flow?

In your conclusions, you say 'At the regional level, ??2.5 concentration was high in areas surrounding Mega cities, and livability was low' (L#30, 31). How can you support this conclusion if you have one data per city? You do not say anything about their surroundings.

Author Response

We are happy to respond to your comments and submit a revised manuscript. We appreciate your suggestions and comments. The corrections were reflected in revised version. The part modified by the comment was marked in red. Comments and answers are indicated by inserting memo into the manuscript. Once again, we sincerely thank you for reviewing and giving advice on our article.

  1. You have improved your paper. However, it does not have statistical soundness: using just one data per year and town is not statistically valid.
  • We used 27 independent variables (livability indicators) in this study and used annual data for 5 years from 2015 to 2019. Therefore, the one data per year and town pointed out by the reviewer is not well understood. We used 27 variables in 5-year intervals in the GRA analysis, and 13 variables with correlation coefficients greater than 0.6 were derived, and panel regression analysis and GIS analysis were performed. In other words, we did not use 1 data per year. Panel data was constructed for spatial and temporal analysis by region.

  1. In your paper still appears y to indicate different variables. That makes a confusion to the reader.
  • (p8, 267-269) As you pointed out, the y variable can be confusing for readers to understand, so we have corrected it.
  •  
  1. In my comment about normalization, you replied:

'In data normalization, the division of expressions into positive data and negative data considers the negative impact on livability in the case of vulnerability and disaster indicators. For exponentiation, it was calculated considering the negative influence.

PM2.5 concentration data was not standardized and was used as it is.'

But this is not explained in the text. So readers can be confused (eq. 1, 2 3 in lines 219-221)

  • As you said, we additionally described what is needed in the data standardization section.

(p6, 218-222) In data normalization, the division of expressions into positive data and negative data considers the negative impact on livability in the case of vulnerability and disaster indicators. For exponentiation, it was calculated considering the negative influence. In case of  concentration data was not standardized and was used as it is.

  1. In your conclusions, you say you have proved L#8 'according to the geographical and social characteristics of the country' Well, I have not been able to see any geographic map, with mountains, rivers.... How does the air flow?
  • (p3, 99-100) In the research purpose of the introduction on page 3, it was described as 'according to the geographical and social characteristics of the country', but there seems to be a misunderstanding in the use of the term. Social characteristics meant that social vulnerability indicators were used, and the word 'geographic' meant spatial analysis. Therefore, confusing terms (geographical and social characteristics) have been removed from the conclusion section and corrected in the introduction section.

  1. In your conclusions, you say 'At the regional level, ??5 concentration was high in areas surrounding Mega cities, and livability was low' (L#30, 31). How can you support this conclusion if you have one data per city? You do not say anything about their surroundings.
  • We were able to examine the spatial and temporal changes of PM2.5 concentration and livability index in the spatial analysis result. The part marked in red is the interpretation of the region. Rather than just one indicator, it was analyzed by indexing 13 variables that measure livability. Although it is difficult to interpret concretely according to the indicators, it is a representative method of measuring the quality of a livable environment.

International organizations such as EIU, Mercer, Monocle, and PwC index the livability indicators every year and announce the rankings of 'livable cities' in the world. We also conducted a comparative analysis of the regions of China and South Korea by using the livability index.

Of course, as you said, there may be limitations in interpreting a city as a livability index. For this reason, in future research, finding on the specific causal relationship of why the regional differences occur should be conducted.

In addition, the fact that Korea, which is close to China, which emits high PM2.5 globally, was the subject of study, it will be able to contribute to joint research between the two countries in the future.

We sincerely thank you for reviewing and giving advice on our article. 

Thank you.

This manuscript is a resubmission of an earlier submission. The following is a list of the peer review reports and author responses from that submission.

Round 1

Reviewer 1 Report

This study attempted to establish the relationship between PM2.5 and urban livability and concluded that air pollution has a negative impact on livability. The subject of the manuscript is interesting. However, in my opinion, there are many major weaknesses that make the manuscript unfit for publication at its current stage. One of my major concerns is that the authors mainly used correlation to conclude that PM2.5 is affecting livability. This is invalid because correlation and causation are different things. This manuscript should provide more robust statistical analyses to make a strong conclusion like the one stated in its title.

Some additional comments are as follows:

1.      Without line numbers, it is very inconvenient to communicate the errors and corrections. Is it the journal’s requirement or the authors’ choice not to put the line numbers? Please use line numbers in future submissions so that reviewers do not need to spend time locating and communicating the errors and issues in your manuscript.

2.      Introduction; the first sentence needs citation.

3.      Introduction, the second sentence is unclear. Do you mean 70% of all-cause deaths? Please specify and also cite the source.

4.      Introduction (Line 9-13): Why did the authors highlight PM2.5 as the transboundary pollutant and also as a long-range transport issue? Actually, there are other pollutants, such as POPs, that are more important in long-range transport. This needs to be explained in the manuscript.

5.      The introduction section is too long. Please make it concise.

6.      Figure 2 quality is poor. The words on the map are barely readable. Please provide a high-resolution map with readable details. Else the map is useless here. 

7.      Section 2.2.1: Please provide the name of the GIS software.

8.      Section 2.3: Please provide the formula used for calculating the relationship coefficient, R. Also, is there any measure of the statistical significance of R?

9.      What is the reason for using the relationship coefficient, R, rather than other common correlation coefficients? Please explain in Section 2.3.

10.  What is the reason for using 2015-2019 for the analysis? Please explain in the manuscript.

11.  Figure 3: Provide axis titles. Also, the text within the figure is barely readable. Same in Figure 4. Make figures readable.

12.  Figures 3 and 4 and Table 2 present the same data. This is unnecessary. Either keep the figures or the table. I suggest removing Figures 3 and 4 as they do not provide any new information not included in Table 2.

13.  Section 3.1. The authors discuss the relationship between PM2.5 and domain, indicators, and sub-indicators. However, they do not provide the meaning of those relationships. For example, what does it mean to have R equal to 0.5653 between PM2.5 and vulnerability and sensitivity for Korea? Similarly, the authors mentioned that “in South Korea, the urban living infrastructure and safety domains were highly correlated”; (the authors meant these are correlated with PM2.5 if I understood correctly)? What is the real-world interpretation of these relationships?

14.  The authors used the term “correlation” and their “R degree” interchangeably. Are they the same? If not, please be careful while using these terms. This makes the manuscript confusing for the readers.

15.  Figure numbering is incorrect. Please get your manuscript proofread before submitting it to journals for peer review.

16.  Page 11, second paragraph, first sentence: “There were significant spatial differences in the annual average ??2.5 pollution…” Did the authors perform a significance test? Where are those results?

17.  Page 12 (Figure): How did the authors interpolate the PM2.5 data over the study area? Which GIS tool and what interpolation technique was used? What is the accuracy of interpolated concentrations? Did the authors validate their results? These important details are lacking in the manuscript.

18.  None of the figures in this manuscript has readable text. The figure resolution is too poor.

19.  Section 4. Discussion: The authors said that “the high correlation between urban living infrastructure and the ??2.5 concentration indicates that the latter clearly affects the quality of the living environment.” The authors should be cautious that “correlation” and “causation” are different. Correlation does not mean causation. Therefore, the authors’ interpretation of this relationship cannot valid in the strict sense.

Reviewer 2 Report

 1. Why did authors use data from 2015 to 2019? In fact, the latest data from 2020 to 2022 have been updated on the government website.

2. Shaanxi, Henan, Anhui, Hubei, Hunan, and Jiangxi are not coastal provinces.

3. Please explain the objectivity, scientificity and rationality of Livability indicators in table 1.

4. Can these vulnerability indicators (X1:Percentage of the population over 65 years; X2: Percentage of the population under 13 years; X3: Ratio of recipients of basic living support; X4: Mortality rate per 100,000 population; X5: Aging index) truly assess the true state of the two countries ? Actually, Population index is not enough to evaluate vulnerability,and there are many factors influencing mortality, which cannot be directly related to air pollution. Furthermore, Aging indicator is only one indicator of population vulnerability.

5. How can authors prove that the impact of these indicators (X6: Urban park area per 1,000 population; X7: Number of students per teacher; X8: Number of childcare facilities per 1,000 children;  X9: Number of employees in medical institutions per 1,000 population X10: Hospital beds per 1,000 population) on urban livability is associated with PM2.5 pollution?

6. What is the relationship between natural disaster damage and air pollution ? How this correlation affects urban urban livability?

Reviewer 3 Report

Thank you for giving me this opportunity to read the manuscript entitled "Air pollution (PM2.5) negatively affects urban livability in coastal cities in South Korea and China". The topic of this manuscript is interesting and would be a good contribution to this field. I think it could be considered for publication in International Journal of Environmental Research and Public Health once the following issues are addressed.

  1. Please replace the keywords that already appear in the manuscript's title with close synonyms or other keywords, which will also facilitate your paper to be searched by potential readers.

  1. I suggest that authors add line numbers to their drafts so that it is easier for reviewers to raise comments.

  1. Some newly published papers regarding PM2.5 emission could be cited in the first paragraph of the Introduction for the statement “As China emits the most PM2.5, measures to reduce PM2.5 pollution are urgently needed (Ju, 2017).”, for example, the paper titled “Dynamic assessment of PM2.5 exposure and health risk using remote sensing and geo-spatial big data.”

  1. Correlation coefficients are suggested to retain to three decimal places.

  1. The consistency of statistical methods in the statistics of two countries, China, and Korea, needs to be discussed, especially for the Chinese data, which are from different statistical departments.

  1. Authors need to use professional formula editing software to edit formulas, and using screenshots may not be acceptable.

  1. A compass should be added to Figure 2.

  1. Some grammatical errors exist in the manuscript. Therefore, a critical review of the manuscript language will improve readability.

Reviewer 4 Report

Dear Authors

topic is interesting, practical value obvious but quality of your presentation need to be improved.

First manuscript is not prepared according to journal requirements. So my comments cannot be as specific as they should be (I wont count lines in text). That is first thing to do.

Second (connected to first one) references are written in text and in reference part not according to journal requirements. Second thing to be fixed.

Third, in many places in text you divide words e.g. "pro-blem" instead of "problem". Please do careful proof reading.

Fourthly, it is a mess in figure numbers (there are two figure 1) and in text first appear figure then mentioning of it - it should be other way round. Please correct.

And fifth, but most important, discussion must be rewrite. It is unimaginable to cite only two articles in discussion chapter. When you do this part again conclusions will need reconsidering.